# In Vitro Fermentation and Chemical Characteristics of Mediterranean By-Products for Swine Nutrition

**DOI:** 10.3390/ani9080556

**Published:** 2019-08-14

**Authors:** Alessandro Vastolo, Serena Calabró, Luigi Liotta, Nadia Musco, Ambra Rita Di Rosa, Monica Isabella Cutrignelli, Biagina Chiofalo

**Affiliations:** 1Department of Veterinary Medicine and Animal Production, University of Napoli Federico II, 80137 Napoli, Italy; 2Department of Veterinary Science, University of Messina, 98122 Messina, Italy

**Keywords:** citrus fruit pulp, olive oil refusal, in vitro fermentation, pigs’ faeces

## Abstract

**Simple Summary:**

By-products are residues obtained from agriculture and/or industrial processes, considered as wastes that could be used in animal nutrition. They are considered valid instruments to reduce feeding costs and breeding environmental impact. These residues may contain beneficial molecules that could be naturally transferred to animal products. In this study, the nutritional characteristics of eight by-products derived from citrus fruit juice (three pulps and two molasses) and olive oil (three olive cake) processes are evaluated for their possible use in pig diets. The chemical composition and fermentation parameters are different when comparing the citrus fruits and olive oil by-products. The citrus by-products are rich in fermentable carbohydrates, while olive oil by-products are rich in un-fermentable carbohydrates and fat. In any case, all the by-products categories show interesting nutritional characteristics. By-products typical of the Mediterranean area could be use in pig nutrition and could be considered an effective system to reduce animal production costs and limit the environmental impact of some production systems.

**Abstract:**

The purpose of the study is to determine the nutritional characteristics of some by-products derived from fruit juice and olive oil production to evaluate their use in pig nutrition. Five by-products of citrus fruit (three citrus fruit pulp and two molasses) and three by-products of olive oil (olive cake) obtained by different varieties are analysed for chemical composition. The fermentation characteristics are evaluated in vitro using the gas production technique with swine faecal inoculum. All the citrus by-products are highly fermentable, producing gas and a high amount of short-chain fatty acids. The fermentation kinetics vary when comparing pulps and molasses. Citrus fruit pulps show lower and slower fermentation rates than molasses. The olive oil by-products, compared to citrus fruits ones, are richer in NDF and ADL. These characteristics negatively affect all the fermentation parameters. Therefore, the high concentration of fiber and lipids represents a key aspect in the nutrition of fattening pigs. The preliminary results obtained in this study confirm that the use of by-products in pig nutrition could represent a valid opportunity the reduce the livestock economic cost and environmental impact.

## 1. Introduction

The EU is the largest pig meat producer and exporter in the world [1]. Nevertheless, the increasing intensification of pig production raises concerns about its sustainability, especially in terms of nutrient sources. Nearly two thirds of the EU’s cereals are used in animal feeds [2], but only an average of 25–30% of global animal dietary gross energy is retained in meat and milk products. Consequently, the relevant proportion of nitrogen and organic matter intake excreted leads to a potentially important environmental impact. According to the European Environment Agency [3], pig slurries in the EU are responsible for about 15% of ammonia (NH_3_) and 4% of total methane (CH_4_) emissions. In the future, the feed industry will need to find alternative feedstuffs and minimize their eco-footprint [4].

In the last few years, there has been widespread social and environmental pressure for the efficient reutilization of agricultural industry residues [5] due to the global intensification of food production, which creates large quantities of food co-products and wastes [6]. Utilization of agro-industrial by-products in farm animal nutrition reduces feeding cost as well as the environmental impact of the food production. Moreover, this kind of recycling improves agriculture profitability, turning low-quality materials into high-quality foods [7].

By-products are not to be considered waste, but raw material obtained from agriculture and/or industrial process [8], and this is compliant with current legislation that strongly encourages the food industry to find new end-uses for refusals [9]. In recent years, European legislators have enacted rules to distinguish waste from by-products [10,11]. In addition, in the last decade, industrial by-products, co-products, insect materials, seaweed ingredients [12,13], and ex-food or former food products have been proposed as categories with great potential as alternative ingredients for animal diets [14,15,16]. Furthermore, these agricultural industry residues pose increasing disposal and potentially severe pollution problems and represent a loss of valuable biomass and nutrients [17]. In addition, industrial ecology and circular economy are considered the leading principles for eco-innovation focusing on a ‘zero waste’ society [18]. The main parameters affecting the extensive application of by-products such as functional feed ingredients in livestock nutrition are related to animal factors, logistics, and their commercial value [19].

The by-products are considered valid instruments to reduce dietary costs, since they contain the feed price variation and benefit from their specific nutritional characteristics. Different studies have underlined the advantage in using residues in poly- and monogastric animal nutrition. Several studies [20,21,22,23] have shown that by-products administration of livestock could improve meat, milk, and cheese nutritional quality. In addition, several by-products contain bioactive substances, called nutraceuticals [24]. Given this, the functional properties of several plant extracts have been investigated for their potential use as novel nutraceuticals; these biological substances are ‘pharmacological multitaskers’ [25]. Research into the use of natural antioxidant and health-promoting compounds from plant sources has resulted in experimental feeding trials, which have examined the effects of plant extracts/nutraceuticals in the diet of dairy and meat-producing animals [26,27,28], derived from their capacity to improve animal health and the quality and nutritional value of food due to the content of bioactive compounds, which may be considered ‘natural functional ingredients’. 

The by-products have high potential for feeding, though their use has different limits due to the large variability in their chemical composition and physical status [8], due to their seasonal production and to their short shelf-life, due to their high moisture and fat levels. To increase the by-products’ shelf-life, some precautions could be adopted such as treating by-products rich in moisture with a hygroscopic substance (e.g., lime) or adding anti-oxidants to the by-products rich in fats.

The production of olive oil and citrus fruits has been widespread in the Mediterranean area from ancient times. As reported by Azbar et al. [29], olives have been cultivated in this area for more than 7000 years, whereas citrus fruits were first imported to the Mediterranean Countries during the tenth century. Both crops play important social and economic roles in Italy, particularly in the south. Italy is the second largest olive oil producing country [30] in the world, producing more than 429,000 tons per year [31], whereas it is the eighth biggest producer of citrus fruits, producing 3.2 million tons per year [32]. This production is generally concentrated in several medium-large cooperatives. Olives and citrus fruits are processed to obtain oil and juice, with the consequent production of enormous quantities of by-products from the industrial process. These by-products represent important nutritional resources [33] thanks to their high nutritional value and bioactive compound levels [22,34,35,36,37].

Many agro-industrial residues, such as citrus pulps, citrus molasses, and olive dry pomace, have the potential to be used to feed livestock animals (ruminants, swine), and several studies have highlighted the presence of nutraceutical components [38,39,40,41].

The catalogue of raw materials for feeds [11] includes the ‘dried citrus pulp’ as ‘product obtained from the citrus fruits processing and subsequently dried’. Its use in animal nutrition has long been known [19], and several studies have confirmed its potential usefulness, mainly concerning its integration in lambs’ [42,43] and pigs [44] diets.

The purpose of the present study is to characterize juice and oil industry by-products, in terms of their chemical composition and in vitro fermentation characteristics, for their possible use in pigs’ diets. With this aim in mind, the in vitro gas production technique is used with pig faecal *inoculum*.

## 2. Materials and Methods

Eight by-products derived from citrus fruit or olive oil processing were evaluated, i.e., dried golden orange pulp (GOP), dried red orange pulp (ROP), dried lemon pulp (LP), concentrated lemon molasses (LM), concentrated orange molasses (OM), dried olive cake variety *Nocellara* (DPN), dried olive cake variety *Biancolilla* (DPB), and dried olive cake variety *Cerasuola* (DPC). All citrus fruit were treated with lime, to reduce the moisture and increase their shelf-life during the storage. On the other hand, olive oil by-products were stabilized by the addition of citric acid. Each by-product was sampled by pooling three different lots of production. 

### 2.1. Chemical Composition

At the laboratory of the Consortium of Meat Research (CoRFilCarni, Messina, Italy), each sample was milled (1.1 mm) and analysed for dry matter (DM), crude protein (CP), ether extract (EE), crude fiber (CF), ash, and starch contents according to AOAC [45] procedures (ID number: 2001.12, 978.04, 920.39, 978.10, 930.05, and 996.11, respectively). Neutral detergent fiber (NDF), acid detergent fiber (ADF), and acid detergent lignin (ADL) were also determined according to Van Soest et al. [46]. 

The analysis of each parameter was replicated three times for each by-product, and the results were reported as a mean of the three replications.

### 2.2. In Vitro Gas Production

The in vitro gas production technique proposed by Theodorou et al. [47] was used to evaluate the fermentation kinetics in a pig’s large intestine. This technique, originally designed to evaluate feedstuffs for ruminant species, was subsequently also utilized in monogastric species using faecal inoculum [48,49,50,51]. At the laboratory of Feedstuffs Evaluation (University of Napoli Federico II, Italy), in vitro gas production trial was performed. Each sample was weighed in triplicate (0.5068 ± 0.0045 g), and 82 mL of anaerobic medium was added into serum flasks. In order to correct fermentation parameters, three bottles were incubated without substrate and were used as blank. As *inoculum*, faecal samples were collected per rectum from three adult neutered Large White pigs (mean age: 350 ± 10 days; mean live weight:159.8 ± 5.2 kg) fed a commercial diet (CP: 14.8%; CF: 4.0%). The faecal pool was filtered through a double thickness of cheesecloth, diluted (1:6) in NaCl solution, homogenized, and added to each bottle (5 mL) under anaerobic conditions. The flasks were incubated at 39 °C for 72 h. During the incubation, the volume and pressure of gas produced were measured every 2-4 h (totally 26 times) using a manual pressure transducer (Cole and Parmer Instrument Co., Vernon Hills, IL, USA). The cumulative gas volume (OMCV, mL/g) was related to the incubated organic matter, and the gas profiles were fitted to the equation described by Groot et al. [52] as follows:$$G=\frac{A}{1+\frac{C^{B}}{t^{B}}}$$
where G is the total gas produced (mL/g of incubated OM) at t (h) time, A is the asymptotic gas production (mL/g of incubated OM), B is the time at which half of the asymptote is reached (h), and C is the switching characteristic of the curve.

The maximum fermentation rate (R_max_, mL/h) and time at which it occurred (T_max_, h) were also calculated [53] as follows:$$R_{max}=\;\frac{A\cdot B^{C}\cdot B\cdot T_{max}{}^{\left(B-1\right)}}{{\left(1+C^{B}\cdot T_{max}{}^{-B}\right)}^{2}}$$
$$T_{max}=C\cdot {\left[\left(B-1)/(B+1\right)\right]}^{\frac{1}{B}}$$
The fermentation was stopped at 4 °C, and the pH of each flask was measured using a pHmeter (model 720A+ Thermo Fisher Scientific, Rodano, MI, Italy). The organic matter disappearance (OMD, %) was determined by filtering the residues under vacuum using pre-weigh sintered glass crucibles (Scott Duran, porosity#2). 

### 2.3. End Products 

Fermentation liquor was collected from each flask after 72 h of incubation in order to determine the ammonia and short-chain fatty acid production. The ammonia (NH_3_, mmol/g) was measured using a spectrophotometer (model Helios gamma UV/VIS Thermo Fisher Scentific, Rodano, MI, Italy), at a wavelength of 623 nm. 

The short-chain fatty acids (SCFA, mmol/g) were determined by centrifuging the samples twice at 12000 × g for 10 min at 4 °C, then 1 mL of supernatant was diluted in 1 mL of oxalic acid 0.06 M. The obtained samples were injected into a gas chromatograph (Thermo Fisher Scientfic, Rodano, MI, Italy; model trace 1310) equipped with a fused silica capillary column (Supelco, 30 m × 0.25 mm × 0.25 μm film thickness), and each fatty acid (acetate, propionate, iso-butyrate, butyrate, valerate and iso-valerate) concentration was measured, comparing the sample peak areas of each SCFA with that of an external standard [54]. Branched-chain fatty acids proportion (BCFA) were also calculated as follows: [(iso-valeriate + iso-butyrate)/SCFA].

### 2.4. Statistical Analysis

Fermentation characteristics (OMCV, OMD, pH), model parameters (T_max_, R_max_), and the final products (SCFA, BCFA, NH_3_) were statistically analysed to detect the differences between substrates by ANOVA for one-way (JMP^®^, Version 14 SW, SAS Institute Inc., Cary, NC, USA, 1989–2019) according to the following model:
y_ij_ = μ + Sub_i_ + ε_ij_
where y is the experimental data, μ represents the general mean, Sub is the substrates (i = 1, 2, ... 8), and ε is the error term. The significance level was verified using HSD Tukey test at *p* < 0.01 and *p* < 0.05.

## 3. Results

In Table 1, the chemical composition of the tested by-products is reported. Most samples had high level of dry matter (around 96.0%), except orange and lemon molasses, where low DM levels were found (52.43 and 44.49%, respectively). The tested by-products had low protein content (varying from 5.36 to 9.14% DM, in OM and DPN, respectively). Regarding lipid content, the three olive oil by-products had more than 29% DM, while the orange fruit by-products were characterized by low fat values, except for lemon pulp (4.89% DM). The structural and reserve carbohydrates content was almost variable among the analysed by-products. Both molasses samples had very low structural carbohydrates values and an untraceable content of starch; consequently, they were composed of 60 % soluble sugars, together with organic acids (6–17% DM), pectins (6–10% DM), and flavonoids (0.5–1.7% DM). The NDF level of citrus pulp by-products was higher than 30% DM, and mainly constituted by cellulose and hemicelluloses, while the three dried pomaces were highly lignified. Very low content of starch was detected in all the by-products, while the highest percentage was found in ROP (2.91% DM). The three citrus pulp samples had a particularly high ash content (mean 17.77% DM ± 0.6), as was the case in lemon molasses. The ash content of the other by-products varied from 3.67 to 4.43 % DM.

The in vitro fermentation characteristics are reported in Table 2. For all the parameters, statistical and significant differences (*p* < 0.01) emerged. Citrus fruit by-products were significantly higher fermentable than the olive oil by-products, as demonstrated by the highest OMD and OMCV values (>70% and >200 mL/g, respectively).

On the other hand, the incubation of dried pomace substrates showed low fermentability and produced a low gas amount. As regards the kinetics parameters, the by-products have been significantly (*p* < 0.01) differentiated into three categories, i.e., the citrus pulp, that showed the intermediate R_max,_ and the highest T_max_, the molasses, that presented the highest fermentation rate that was reached in the lowest time, and the olive oil by-products, which showed the lowest rate of fermentation and needed more than eight hours to reach it. To better highlight the differences between the by-product categories and to understand the curve trends, the fermentation rates are depicted in Figure 1.

In vitro fermentation end-products of fermenting liquor are reported in Table 3. Significant (*p* < 0.01) differences were observed for pH values registered at the end of fermentation; DPB had the highest value and ROP the lowest. The molasses samples showed the highest (*p* < 0.01) SCFA production (mean values:165.5 mmol/g ±14.1), followed by both the orange fruit pulps (mean values: 107.2 mmol/g ± 9.2); very low SCFA production was detected in the three pomaces (mean values: 43.11 mmol/g ± 2.9). Acetate and propionate were the most representative fatty acids of all the substrates; they represented more than 75% of the total SCFA. The fermentation of all the citrus fruit by-products and, in particular, of both molasses resulted in there being high proportion of butyrate (18 and 13% SCFA, for molasses and pulps, respectively). On the other hand, BCFA production was higher for oil olive by-products than all the citrus fruit by-products.

Ammonia production was significantly higher for both molasses samples, while the DPN showed the significantly lowest values.

The in vitro fermentation characteristics are reported in Table 2. For all the parameters, statistical and significant differences (*p* < 0.01) emerged. Citrus fruit by-products resulted in being significantly higher fermentable than the olive oil by-products, showing the highest OMD and OMCV values (>70% and >200 mL/g, respectively).

## 4. Discussion

### 4.1. Citrus Fruit By-Products

Overall, the chemical composition data of citrus fruit by-products were in agreement with those reported in literature [33,44], considering the differences in terms of production process as well as botanical varieties. As expected, pulp and molasses, coming from the same production process, were consistently different due to their physical and chemical characteristics. The pulps, which made up the waste solid part, were rich in dry matter and fermentable carbohydrates, represented by soluble dietary fiber, such as pectin, an interesting alternative and a low-priced source of fiber in animal feed. Citrus pectin has multiple biological activities, including glycaemic and cholesterol level control [55,56]. Conversely, molasses is the liquid waste of juice production process and, consequently, compared to the pulps presented lower concentration of all the nutrients. The chemical composition of these by-products was reflected in their specific fermentation characteristics. In fact, all the citrus by-products were highly fermentable, producing a high amount of gas and short-chain fatty acids, particularly acetate, propionate, and butyrate. Despite this, the specific fermentation kinetics varied when comparing pulps and molasses. The citrus fruits pulp, richer in complex fermentable carbohydrates, showed significantly lower and slower fermentation rates than molasses, suggesting that they are more fermented into the distal part of gastrointestinal tract [50] and could be considered as pre-biotics. Instead, the two molasses were richer in sugars and showed a shorter and more intense fermentation process; thus, they are mainly solubilised into the stomach and small intestine and may represent a useful energy source. In any case, the pH values respected the physiological range (5.5–7.5) reported by Younes et al. [57]. The production of short-chain fatty acids was significantly high for all the citrus fruit by-products. In particular, the high butyrate production indicates a potential pre-biotic rule of these substrates [58]. The fermentable carbohydrates represented an optimal pabulum for microbiota growth, and the high proportion of butyrate could be useful for the colonic epithelium as a main energy source for cell growth and differentiation [58]. In pulps, the high concentration of ash, due to the lime treatment, reduced the fermentation processes.

As demonstrated by previous in vivo studies [59], the use of citrus fruit by-products in partial substitution of hay to feed Nero Siciliano pigs did not affect swine performance and meat quality in terms of oxidative stability, protein, and fat content. Moreover, they improved the fatty acid profile, reducing significantly saturated fatty acids amount and increasing polyunsaturated fatty acids of n-3 and n-6 series levels. Lemon by-product administration also affected positively the sensorial characteristics of Nebrodi cured sausages due to the volatile fraction of citrus essential oils, which were transferred from citrus to adipose tissue [60,61,62].

### 4.2. Olive Oil By-Products

The chemical composition of olive cake varies widely depending on the olive variety, the proportion of its main components (skin, pulp, and stone), and the oil extraction process [63]. For this reason, it was considered important to characterize the individual by-product, differentiating it according to the most represented varieties in Sicily (*Nocellara*, *Biancolilla*, and *Cerasuola*). As reported by Molina-Alcaide et al. [64], all pomace samples, the residue obtained after the pressure extraction of oil from the entire olive fruit, showed a high percentage of ether extract. These substrates were also characterized by high levels of NDF, which had high lignin content, whereas the level of crude protein was quite low. As a consequence, the substrates were less fermentable, as demonstrated by all the fermentation parameters and kinetics profile. Despite that, the good proportion of butyrate on the total SCFA (always higher than 10%) was indicative of a partial bacterial utilization [54]. As regards the fiber levels, during the last 15 years, efforts have been made to formulate diets that better meet the pig’s requirements to improve performance and to contribute to reducing cost, odour, and pollutant excretion. Dietary fiber in pig reduced ammonia emission; improved gut health; and, consequently, the pig welfare [65,66]. The dietary fiber escaping digestion in the upper part of the gastro-intestinal tract was potentially available for bacterial fermentation in the large intestine. This means that the gut microflora of healthy animals could be modified in relation to the presence of fiber in the diet [67]. Dietary fiber significantly modifies the microbial equilibrium with a positive or a detrimental impact on animal health, according to the dietary fiber source and the physiological status of the pig. Chiofalo et al. [60] reported that the high-fiber diet caused a negative effect on some pig performances, even if the technical and nutritional characteristics of the meat were not influenced by the dietary treatment. Despite the mentioned disadvantages, there have been several important advantages that are inexpensive protein sources and can be grown at small-farm level. Often, they are by-products or co-products of multipurpose crops. Further studies devoted to the relationship between dietary fiber content and its functionalities are necessary to identify an appropriate fiber level that reduces ammonia emission, promotes intestinal health, and still allows acceptable pig growth rates.

The digestive utilization of these by-products is probably reduced by high lignin concentration, albeit by the fact that the high percentage of fat that these by-products produce could represent an interesting characteristic. Lipids, in fact, supply a greater energy yield compared to carbohydrates and proteins. From a biochemical point of view, fats contain carbon and hydrogen in a more reduced state; therefore, these elements can potentially be much more oxidised [14]. For this reason, the supplementation of fat in swine nutrition is a common practise. In addition, the proportion in monounsaturated fatty acids (MUFA), especially in oleic acid, could turn it into an interesting dietary ingredient because it might modify the fatty acid profile of the pig fat tissues [68,69]. The increasing demand from modern society for healthy meat has to be taken into account. A meta-analysis of epidemiological studies has cast doubt on the relationship between long chain SFA and cardiovascular diseases [70]. Instead, the increase of the MUFA percentage in relation to the olive oil by-products intake modifies the unsaturation degree and the atherogenic index. It has been shown that oleic acid (cis-9-octadecanoic acid) has beneficial effects on blood cholesterol and other health related outcomes in humans. Moreover, despite the fact that the cholesterol-lowering response to PUFA is greater than that to MUFA, the potentially adverse health effects of lipoperoxidation products are higher for PUFA in relation to the higher presence of double bonds.

## 5. Conclusions

The preliminary results obtained in this study confirm that the use of local by-products in pig nutrition could represent a valid system to reduce the animal production cost and limit the environmental impact of some production systems typical of the Mediterranean area.

Chemical data, associated with the supplementary information obtained from the in vitro gas fermentation technique (fermentation characteristics, kinetics, and end-products), allow us to estimate the potential utilization along the digestive tube.

Results allow one to consider the by-products analysed in relation to their specific nutritional characteristic for different uses in swine nutrition. Whereas citrus fruit pulp could improve microbiota and enterocytes homeostasis, citrus fruit molasses could probably increase the diet nutritional values due to its high concentration in soluble sugars. A different approach is necessary in the evaluation of dried olive cake, which has interesting nutritional characteristics, but its utilization is prevented by its high lignin concentration.

Further studies could be performed focusing on the effect of raw material processing conditions on the composition (characterization and quantification of the active/target compounds) in the produced by-products, as well as evaluating the in vitro enzymatic digestibility that occurs in the proximal tract of the digestive tract and identifying the bioactive compounds and/or the toxic factors for the better exploitation of these residual biomasses in the agroindustry. This will allow us to take into consideration the citrus fruit and olive oil by-products as a food resource and not as a waste, whose correct disposal will always be an added cost for the food companies, and to evaluate the functionality of new types of feed ingredients in large-scale (commercial) feeding trials.

## Figures and Tables

**Figure 1 animals-09-00556-f001:**
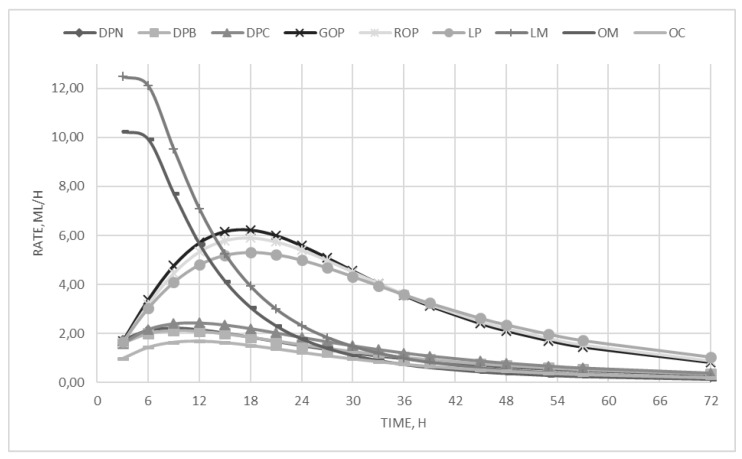
In vitro fermentation rate over time.

**Table 1 animals-09-00556-t001:** Chemical composition.

Samples	DM	CP	EE	NDF	ADF	ADL	Ash	Starch
	%	% DM						
GOP	95.75	5.82	1.10	31.80	23.35	3.74	18.07	2.11
ROP	96.58	6.02	0.91	33.05	25.72	3.69	17.08	2.91
LP	96.90	7.42	4.89	37.42	28.37	4.23	18.16	1.91
LM	44.49	6.74	0.43	1.37	0.36	0.20	15.01	-
OM	52.43	5.36	0.40	0.92	0.27	0.11	7.15	-
DPN	95.63	9.14	29.51	50.19	38.83	27.19	4.43	1.76
DPB	95.66	8.67	30.04	53.88	39.69	21.49	3.67	1.11
DPC	95.55	8.07	31.46	44.05	39.65	20.50	4.17	1.57

GOP: dried golden orange pulp; ROP: dried red orange pulp; LP: dried lemon pulp; LM: concentrated lemon molasses; OM: concentrated orange molasses; DPN: dried olive cake variety *Nocellara*; DPB: dried olive cake variety *Biancolilla*; DPC: dried olive cake variety *Cerasuola*. DM: dry matter; CP: crude protein; EE: ether extract; NDF: neutral detergent fiber; ADF: acid detergent fiber; ADL: acid detergent lignin.

**Table 2 animals-09-00556-t002:** In vitro fermentation characteristics.

Samples	OMD %	OMCV mL/g	T_max_ H	R_max_ mL/h
GOP	81.23^B^	222.2^A^	16.9^A^	6.27^C^
ROP	83.91^B^	218.9^A^	17.3^A^	5.91^C^
LP	72.67^C^	212.6^A^	18.1^A^	5.30^C^
LM	91.82^A^	211.8^A^	4.08^C^	13.0^A^
OM	93.63^A^	167.3^B^	4.06^C^	10.7^B^
DPN	23.47^E^	77.66^C^	8.90^B^	2.21^D^
DPB	24.17^E^	78.78^C^	9.30^B^	2.20^D^
DPC	31.74^D^	89.11^C^	11.1^B^	2.44^D^
RMSE	1.52	9.67	0.73	0.61

OMD: organic matter disappearance; OMVC: cumulative volume of gas related to incubated organic matter; R_max_: maximum fermentation rate; T_max_: time at which R_max_ occurs; RMSE: root mean square error. Along the column, different letters indicate difference for *p* < 0.01.

**Table 3 animals-09-00556-t003:** In vitro fermentation end products.

		Samples								
		GOP	ROP	LP	LM	OM	DPN	DPB	DPC	RMSE
pH		6.71^ab^	6.61^b^	6.65^ab^	6.70^ab^	6.68^ab^	6.75^a^	6.76^a^	6.74^a^	0.018
Ace	mmol/L	68.95^C^	80.53^BCbc^	56.73^Cd^	117.0^Aa^	95.87^ABb^	27.49^D^	23.35^D^	27.20^D^	7.34
Prop	“	17.37^A^	17.70^A^	15.84^A^	19.77^A^	18.59^A^	7.96^B^	7.32^B^	7.92^B^	1.51
Iso-But	“	1.50^BCDab^	1.70^B^	1.57^BCa^	2.70^A^	2.53^A^	1.06^Dc^	1.09^CDc^	1.12^CDbc^	0.13
But	“	12.60^DCa^	13.22^C^	9.48^Db^	23.51^B^	27.95^A^	4.65^E^	4.57^E^	5.26^E^	1.06
Iso-vale	“	2.65^C^	2.47^C^	2.53^C^	4.93^A^	3.85^B^	2.12^C^	2.20^C^	2.33^C^	0.27
Vale	“	1.71^BC^	1.75^BC^	2.18^BCa^	3.78_A_	3.42^A^	1.27^Cb^	1.18^Cb^	1.25^Cb^	0.32
SCFA	“	104.8^BC^	117.4^BCa^	88.35^Cb^	171.7^A^	152.2^A^	44.55^D^	39.71^D^	45.08^D^	8.28
BCFA	“	0.030^B^	0.030^B^	0.040^B^	0.030^B^	0.037^B^	0.063^A^	0.070^A^	0.063^A^	0.006
NH_3_	“	19.44^BC^	21.87^BC^	22.68^BCa^	43.64^A^	42.92^A^	14.96^Cb^	22.87^BCa^	25.68^B^	2.47

Ace: acetate; Prop: propionate; Iso-But: Iso-Butyrate; But: butyrate; Iso-Val: Iso-Valerate; Val: valerate; SCFA: total short-chain fatty acids; BCFA: branched-chain fatty acids; NH_3_: ammonia. RMSE: root mean square error. Along the column, different lowercase letters indicate difference for *p* < 0.05 and different capital letters indicate difference for *p* < 0.1.

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
