# Peer review of "In Vitro Fermentation and Chemical Characteristics of Mediterranean By-Products for Swine Nutrition"

_animals, 2019, doi:10.3390/ani9080556_

Round 1
Reviewer 1 Report
Overall paper needs some editing for English grammar and clarity –best choice of words – wording.
Simple summary – By-products are residues obtained … considered as wastes that could be considered to be used in animal nutrition. (the It could be doesn’t fit – go from plural to singular) back to plural – they –
Sometimes or These residues may contain beneficial molecules
Possible use in pig diets ?
Line 21 characteristics
Don’t need the results suggest –--
Line 30 (gas or more specifically VFA’s ) – do we want pigs to produce methane a greenhouse gas ?
32 fermentation rates (correct – since there were multiple by-products)?
Line 33 negatively affected
What is nutritional key? line 35 –
Line 42 – production expansion in china – how does this relate to the paper and now with ASF – China has greatly reduced production – don’t include temporary “facts” that make paper out of date immediately.
Line 57 to 58 – don’t know if soybean meal is a byproduct – yes soybean oil is removed but really just splitting the soybeans into two products---- and need to process each separately.
Line 79 to 80 – healthy properties – correct wording? (Explained later so it this needed here?)
Line 88 to 89 – the compound could affect the animal – change its chemical composition – fatty acid profile – oxidative status – and the compound not be transferred into the end product effectively – in some cases transfer of the bioactive compounds is desirable and other cases the indirect effect of the bioactive compound to impact the animals health, or chemical composition – oxidative status – animal well-being – even food safety via microbiome – (less AB use) – may be an objective.
Line 95 – what about fat oxidation? ever cost effective to add anti-oxidants ?
Line 101 producing 200,000 tons per year?
103 per year?
Line 142 into – in a pig’s large intestine?
Line 143 monogastric species
Table 1 it seems that for LM and OM – the chemicals listed only account for 60 to 80 % of the total – is it discussed what the major compounds – chemical families of compounds in in the OM and LM?
line 183 Most samples had – except
Line 185 the by-products had low protein content
(don’t say showed or resulted in journal paper – just state the results – the samples had – were – etc
Line 191 – okay --- were soluble sugars a major component of the OM and LM ? (approx. what level and any pectin ? – soluble fiber? ) –
Line 195 – (the text continues here not needed)
Line 154 perhaps give the groot equation in the text and in table 2 –
Line 229 – instead of showed – had or values were –
Line 253 in agreement with –
Line 256 due to their
Line 226 had
Line 276 – affected the fermentability - - not clear
Line 291 highly lignified – had high lignin content –
Line 310 to 306 long run on sentence – 2 sentences??
Line 311 – acceptable pig growth rates.
Line 322 – saturated fat no longer considered bad fat for the past 5 years – sugar is the evil – trans 18:1 is bad and too high of n-6 is an issue.
Line 359 digestive tract
Is there a reference that the in vitro method does show a relationship to actual pig data when different feedstuffs are fed to pigs?
Author Response
Dear Editor
Please find the revised version of the manuscript Animals 560777 entitled “
We addressed all reviewer’s concerns as reported below. We wish to thank the reviewers for useful comments, which allowed us to improve the quality of the paper. Hope the manuscript can be considered for publication in its present form. All changes were underlined.
Reviewer 1
Comments and Suggestions for Authors
Overall paper needs some editing for English grammar and clarity –best choice of words – wording.
Simple summary – By-products are residues obtained … considered as wastes that could be considered to be used in animal nutrition. (the It could be doesn’t fit – go from plural to singular) back to plural – they – DONE
Sometimes or These residues may contain beneficial molecules DONE
Possible use in pig diets? DONE
Line 21 characteristics DONE
Don’t need the results suggest DONE
Line 30 (gas or more specifically VFA’s) – do we want pigs to produce methane a greenhouse gas? Sentence was changed
32 fermentation rates (correct – since there were multiple by-products)? DONE
Line 33 negatively affected DONE
What is nutritional key? line 35 DONE
Line 42 – production expansion in china – how does this relate to the paper and now with ASF – China has greatly reduced production – don’t include temporary “facts” that make paper out of date immediately. DONE
Line 57 to 58 – don’t know if soybean meal is a byproduct – yes soybean oil is removed but really just splitting the soybeans into two products---- and need to process each separately. DONE
Line 79 to 80 – healthy properties – correct wording? (Explained later so it this needed here?) Deleted
Line 88 to 89 – the compound could affect the animal – change its chemical composition – fatty acid profile – oxidative status – and the compound not be transferred into the end product effectively – in some cases transfer of the bioactive compounds is desirable and other cases the indirect effect of the bioactive compound to impact the animals health, or chemical composition – oxidative status – animal well-being – even food safety via microbiome – (less AB use) – may be an objective. The sentence was deleted
Line 95 – what about fat oxidation? ever cost effective to add anti-oxidants? DONE
Line 101 producing 200,000 tons per year? Yes, the productions of olive oil and citrus fruits are per year. WE had updated the olive oil production in 2018
103 per year? DONE
Line 142 into – in a pig’s large intestine? DONE
Line 143 monogastric species DONE
Table 1 it seems that for LM and OM – the chemicals listed only account for 60 to 80 % of the total – is it discussed what the major compounds – chemical families of compounds in in the OM and LM?
Line 183 Most samples had – except. We changed showed with had and excepted with except. The major compounds are discussed into the text (Results)
Line 185 the by-products had low protein content DONE
(don’t say showed or resulted in journal paper – just state the results – the samples had – were – etc
Line 191 – okay --- were soluble sugars a major component of the OM and LM? (approx. what level and any pectin? – soluble fiber?) Yes, the soluble sugars account for approximately 60%; but together with organic acids, pectins and flavonoids account for, respectively, 6 -17% on DM, 6-10% on DM, 0.5-1.7% on DM. DONE
Line 195 – (the text continues here not needed) DONE
Line 154 perhaps give the Groot equation in the text and in table 2 The equations have been inserted into the text
Line 229 – instead of showed – had or values were – DONE
Line 253 in agreement with DONE
Line 256 due to their DONE
Line 226 had DONE
Line 276 – affected the fermentability - - not clear Sentence was changed
Line 291 highly lignified – had high lignin content – DONE
Line 310 to 306 long run on sentence – 2 sentences?? Sentences were changed
Line 311 – acceptable pig growth rates. DONE
Line 322 – saturated fat no longer considered bad fat for the past 5 years – sugar is the evil – trans 18:1 is bad and too high of n-6 is an issue. Yes, we agree with you and we have reformulated the sentence.
Line 359 digestive tract DONE
Is there a reference that the in vitro method does show a relationship to actual pig data when different feedstuffs are fed to pigs?
Several studies Bauer et al. [48], Bindelle et al. J. Anim. Sci 2009, compared the data obtained by in vitro fermentation trials using pig faecal inoculum with that one obtained in vivo. These authors concluded that t in vitro is a useful tool for the formulation of pig diets. In terms of digestibility, nitrogen excretion and short chain fatty acids production.
Reviewer 2 Report
Manuscript entitled „Nutritional evaluation of Mediterranean by-products for swine nutrition” by Vastolo and co-workers cannot be accepted for publication in Animals in the present form. Manuscript needs some changes and explanations. Below you can find my suggestion for consideration:
1. The title is misleading since the paper comprises only chemical composition and in vitro fermentation, but not the nutritional (energy and protein) value.
2. Introduction is very long. Please, give only the main information, which are necessary for clear formulation of working hypothesis and the aim of the study.
3. By-products contain not only nutrients, but also antinutritional factors. Did you analyse them? Nutritional value of by-products depend on type and content of antinutritional factors.
4. In my opinion conclusion is missing.
Author Response
Dear Editor
Please find the revised version of the manuscript Animals 560777
We addressed all reviewer’s concerns as reported below. We wish to thank the reviewers for useful comments, which allowed us to improve the quality of the paper. Hope the manuscript can be considered for publication in its present form. All changes were underlined.
Reviewer 2
Comments and Suggestions for Authors
Manuscript entitled „Nutritional evaluation of Mediterranean by-products for swine nutrition” by Vastolo and co-workers cannot be accepted for publication in Animals in the present form. Manuscript needs some changes and explanations. Below you can find my suggestion for consideration:
1.The title is misleading since the paper comprises only chemical composition and in vitro fermentation, but not the nutritional (energy and protein) value. The title was changed
Introduction is very long. Please, give only the main information, which are necessary for clear formulation of working hypothesis and the aim of the study. The introduction was partially reduced By-products contain not only nutrients, but also antinutritional factors. Did you analyse them? Nutritional value of by-products depends on type and content of antinutritional factors. The sentence was deleted In my opinion conclusion is missing. The paragraph was re-writtenRound 2
Reviewer 2 Report
Manuscript is still not ready for publication in Animals. It should be corrected by Authors.
Table 1: Please, uniform expression of chemical composition. It should be % or % of DM for all.
I did not get any response or comment to my question regarding content of antinutritional factors in by-products.
Please, check conclusion section. It should be corrected. There is statement which should be deleted.
Author Response
Point 1: Table 1: Please, uniform expression of chemical composition. It should be % or % of DM for all.
Response 1: Table 1 was corrected
Point 2: I did not get any response or comment to my question regarding content of antinutritional factors in by-products.
Response 2: We agree with you about the nutritional value of the by-products in relation to their content in antinutritional factors. Unfortunately, in this preliminary note we did not analyse the antinutritional factors in these by-products. We intend to do it in a second note.
Point 3: Please, check conclusion section. It should be corrected. There is statement which should be deleted.
Response 2: The sentence was deleted.